# BENCHMARKING LLMS ON AUTHENTIC CASES FROM MEDICAL JOURNALS

## ABSTRACT

In recent years, large language models (LLMs) have demonstrated remarkable capabilities in the medical domain. However, existing medical benchmarks suffer from performance saturation and are predominantly derived from medical exam questions, which fail to reflect the complexity of real-world clinical scenarios. To bridge this gap, we introduce **ClinBench**, a challenging benchmark based on authentic clinical cases sourced from authoritative medical journals. Each question retains the complete patient information and clinical test results from the original case, effectively simulating real-world clinical practice. Additionally, we implement a rigorous human review process involving medical experts to ensure the quality and reliability of the benchmark. ClinBench supports both **textual and multimodal** evaluation formats, covering 12 medical specialties with over 2,000 questions, including a dedicated rare disease track, providing a comprehensive resource for assessing the medical reasoning capabilities of LLMs. We evaluate the performance of over 20 open-source and proprietary LLMs and benchmark them against human medical experts. Our findings reveal that human experts still retain an advantage within their specialized fields, while LLMs demonstrate superior overall performance on a broader range of medical specialties.

## 1 INTRODUCTION

Recent advancements in large language models (LLMs) have demonstrated a remarkable ability to understand and generate medical content, marking significant progress in the medical field Thirunavukarasu et al. (2023); Liévin et al. (2024); Clusmann et al. (2023); Chen et al. (2024a). Their impressive performance underscores their potential to approach expert-level intelligence.

With the rapid advancement of medical LLMs, existing medical benchmarks lack sufficient challenge and face the issue of performance saturation. For instance, powerful LLMs such as GPT-4o OpenAI (2024), Gemini-2.5-Pro Guo et al. (2025) have achieved accuracy approaching 90% on widely used medical benchmarks like MedQA Jin et al. (2021), MedMCQA Pal et al. (2022) and PubMedQA Jin et al. (2019). To address this limitation, recent works Zuo et al. (2025); Tang et al. (2025); McDuff et al. (2025) have attempted to introduce more challenging benchmarks by incorporating advanced, expert-level examinations, such as medical licensing tests.

However, **these benchmarks remain predominantly exam-oriented and fail to reflect realistic clinical scenarios**. Real-world clinical scenarios require physicians to integrate medical knowledge with practical experience to navigate complex and multifaceted situations, taking into account patient symptoms, medical history, imaging findings, and various diagnostic test results. In contrast, exam questions typically assess isolated pieces of medical knowledge without adequately simulating complex clinical contexts. Consequently, these questions do not sufficiently capture the intricacies and nuances of real-world clinical scenarios.

To address these limitations, we introduce ClinBench: a challenging and real-world medical benchmark for future medical LLMs. Our ClinBench has four key features: (1) **Real-world clinical scenarios:** The questions are sourced from **authoritative medical journals and based directly on real-world clinical cases**. Each question retains the complete patient information and clinical test results from the original case, effectively simulating real-world clinical practice. (2) **High challenge:** The questions are highly challenging, even for experienced physicians. Each question corresponds to a realistic clinical task, requiring specialized medical knowledge, analytical skills,

| Dataset | # Size | # Avg Lens | # Avg Option Num | # Saturation | Real-Med Scenario | Rare-Diseases |
|---------|--------|-----------|------------------|--------------|-------------------|---------------|
| MedQA (USMLE) Jin et al. (2021) | 1,273 | 116.6 | 4 | ✓ | ✗ | ✗ |
| PubMedQA (PQA-L) Jin et al. (2019) | 1,000 | 14.4 | 3 | ✓ | ✓ | ✗ |
| MedMCQA Pal et al. (2022) | 4,183 | 12.8 | 4 | ✓ | ✗ | ✗ |
| MMLU (Med) Hendrycks et al. | 1,089 | 100.1 | 4 | ✓ | ✗ | ✗ |
| MMLU-Pro (Med) Wang et al. (2024) | 586 | 166.6 | 10 | ✗ | ✗ | ✗ |
| MedXpertQA$_{\text{Text}}$ Zuo et al. (2025) | 2,450 | 257.4 | 8.7 | ✗ | ✗ | ✗ |
| HealthBench Arora et al. (2025) | 5000 | - | - | ✗ | ✓ | ✗ |
| ClinBench$_{\text{Text}}$ (Ours) | 2,014 | 462.1 | 8.4 | ✗ | ✓ | ✓ |

Table 1: Comparison with existing textual medical benchmarks. **# Saturation** indicates whether the dataset suffers from performance saturation. **Real-Med Scenario** denotes whether the questions reflect real-world clinical settings. **Rare-Diseases** indicates whether the benchmark includes the evaluation of rare diseases.

| Dataset | # Size | # Images | # Image Rate | # Avg Lens | # Saturation | Real-Med Scenarios | Rare-Diseases |
|---------|--------|----------|--------------|-----------|--------------|--------------------|--------------|
| PMC-VQA Zhang et al. (2023b) | 33,430 | 29,021 | 0.9 | 61.8 | ✓ | ✗ | ✗ |
| OmniMedVQA Hu et al. (2024) | 127,995 | 118,010 | 0.9 | 42.4 | ✓ | ✗ | ✗ |
| GMAI-MMBench Ye et al. (2024) | 21,281 | 21,180 | 1.0 | 49.9 | ✓ | ✗ | ✗ |
| MMMU (H & M) Yue et al. (2024) | 1,752 | 1,994 | 1.1 | 83.6 | ✓ | ✗ | ✗ |
| MMMU-Pro (H & M) Yue et al. (2024) | 346 | 431 | 1.3 | 107.1 | ✗ | ✗ | ✗ |
| MedXpertQA$_{\text{MM}}$ Zuo et al. (2025) | 2,000 | 2,852 | 1.4 | 149.4 | ✗ | ✗ | ✗ |
| ClinBench$_{\text{MM}}$ (Ours) | 2,014 | 4,978 | 2.5 | 421.7 | ✗ | ✓ | ✓ |

Table 2: Comparison with existing multimodal medical benchmarks. **# Image Rate** refers to the average number of images included per question. **# Saturation**, **Real-Med Scenario**, and **Rare-Diseases** are consistent with Table 1.

medical image interpretation, and extensive clinical experience. (3) **Quality assurance:** For each question, the stem is derived from authoritative medical cases. The golden answer is provided by an expert panel, and each question is thoroughly reviewed and validated by human experts. (4) **Comprehensive evaluation:** ClinBench provides both textual and multimodal versions, covering 12 medical specialties and encompassing more than 2,900 questions. Additionally, the inclusion of a dedicated rare-disease track further enhances its clinical comprehensiveness, offering a more comprehensive evaluation.

We evaluate over 20 LLMs, including both open-source and proprietary LLMs. Additionally, we engage attending-level human medical experts to answer ClinBench questions, facilitating a comparison between human experts and LLMs. Our key contributions are summarized as follows:

- We propose **ClinBench**, the first medical multiple-choice benchmark focusing on realistic clinical scenarios. ClinBench has both textual and multimodal versions, with questions derived from authentic clinical cases, closely simulating the real-world scenarios.

- ClinBench is built upon authoritative medical journals with rigorous quality assurance processes. Comprehensive human checks and data leakage risk assessments are conducted to ensure the reliability and quality of the questions.

- We evaluate ClinBench across more than 20 LLMs, providing a comprehensive assessment of the current medical capabilities of existing medical LLMs. Furthermore, through the comparison between human experts and LLMs, we find that human experts still retain an advantage within their specialized fields, while LLMs demonstrate superior overall performance across a broader range of medical specialties.

## 2    COMPARISON WITH EXISTING BENCHMARKS

**Statistic Comparison.** As shown in Tables 1 and 2, traditional text medical benchmarks like MedQA and PubMedQA have short questions with limited options, lacking the challenge of complex, specialized medical tasks. Additionally, MMLU Hendrycks et al., MMLU-pro Wang et al. (2024) and MedXpertQA Zuo et al. (2025) datasets, mostly sourced from educational exams, fail to accurately represent real clinical tasks. In contrast, our ClinBench$_{\text{Text}}$ includes longer, more complex questions with multiple options, all derived from authoritative case journals, offering a better reflection of real clinical scenarios. Moreover, ClinBench$_{\text{MM}}$ incorporates more images per question compared to existing multimodal medical benchmarks, reflecting the complexity of real-world mul-

timodal medical scenarios. The recent emergence of HealthBench Arora et al. (2025) has introduced rubric-based evaluation for open-ended medical tasks, where large language models are required to score responses according to predefined criteria. While such a rubric benchmark is suitable for open-ended assessments, it adds additional complexity to the evaluation process. By contrast, our ClinBench adopts a multiple-choice format, enabling more straightforward, consistent, and reliable evaluation.

**Discriminative Comparison.** As LLMs continue to advance, existing medical benchmarks struggle to effectively evaluate the performance gap among models. As illustrated in Figure 1, compared to benchmarks such as MMLU-Pro and MedQA, our **ClinBench** demonstrates greater discriminative power, enabling clearer distinctions between model performances. This discriminative nature makes ClinBench as a more suitable benchmark for evaluating and guiding the future development of medical LLMs.

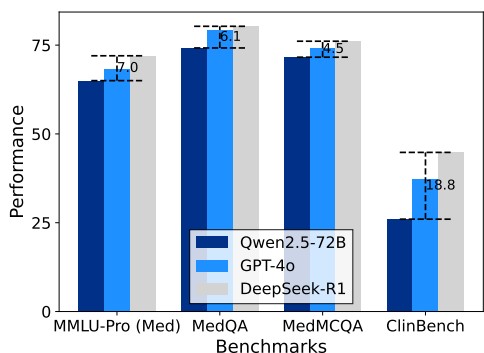

Figure 1: Performance gap among models on different benchmarks. The three exam-oriented benchmarks fail to effectively distinguish capability differences among models, while ClinBench$_{\text{Text}}$ provides a clearer gap.

## 3 THE CLINBENCH BENCHMARK

### 3.1 OVERVIEW

ClinBench consists of two versions: textual and multimodal. The textual version, ClinBench$_{\text{Text}}$, contains 2,014 multiple-choice questions covering 12 distinct medical specialties. The multimodal version, ClinBench$_{\text{MM}}$, is based on ClinBench$_{\text{Text}}$ but partially replaces textual information in question stems with medical images.

The questions in ClinBench are sourced from authoritative medical journals available on PubMed Central[1], a reputable repository of English-language medical case reports. We construct our benchmark based on medical journals for three primary reasons: (1) Medical journals offer comprehensive patient information and detailed clinical test results, thereby closely simulating the real-world diagnostic process. (2) The diagnoses and treatment plans reported in these journals are reviewed and validated by expert medical panels, ensuring the reliability and authority of the reference answers. (3) All patient information in these journals is rigorously anonymized, effectively addressing privacy and ethical concerns. Figure 4 provides two illustrative example from ClinBench$_{\text{Text}}$ and ClinBench$_{\text{MM}}$, demonstrating a challenging question that integrates detailed patient information with medical imaging data.

### 3.2 THE CONSTRUCTION OF CLINBENCH$_{\text{TEXT}}$

Figure 2 illustrates the construction process of ClinBench$_{\text{Text}}$, which consists of three steps: data preprocessing, question stem construction, and candidate options construction.

**Data Preprocess.** We first convert medical journal PDFs into text format using the MinerU tool[2]. Then, we apply a three-step filtering pipeline to the medical journals: (1) **Filtering for Diagnostic Cases:** We first select medical cases from PubMed Central (32M) using keyword tags. As a result, we obtain the full text of approximately 40K English medical case journals. (2) **Filtering for Complete Cases:** We then apply rule-based filtering to exclude incomplete case reports, retaining only those that contain essential sections: patient information, clinical test results, diagnostic conclusions, and treatment plans. Additionally, we discard cases that lack medical images in the patient information and clinical test results, ensuring that each question includes medical images. After this step, we obtain around 4K high-quality, complete medical cases. (3) **Removing Duplicates:**

---

[1]https://pmc.ncbi.nlm.nih.gov/
[2]https://mineru.net/

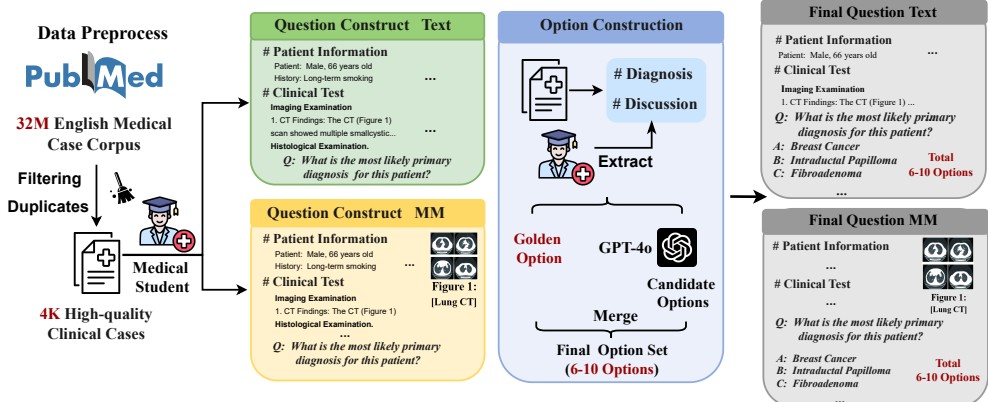

Figure 2: The pipeline for constructing ClinBench. ClinBench$_{MM}$ is built upon ClinBench$_{Text}$, with the difference being that the question stems in the multimodal version have the textual descriptions of the associated medical images removed.

Finally, we eliminate duplicate or highly similar cases to maintain the dataset's diversity and quality. A more detailed process is shown in Appendix C.

**Question Stem Construction.** ClinBench focuses on diagnostic problems because they represent a core, high-stakes clinical challenge that is amenable to objective, verifiable evaluation. This stands in stark contrast to tasks like treatment planning, whose inherent flexibility and open-ended nature render their assessment less standardized. To construct the questions, we recruited 30 undergraduate medical students who followed our detailed annotation guidelines (see Appendix C). For each case, they extracted the *Patient Information* and *Clinical Test Results* sections from the source material to form the question stem. Finally, every question in the dataset is uniformly framed as: "What is the most likely primary diagnosis for this patient?"

**Candidate Options Construction.** The construction of candidate options proceeds in four main sub-steps (see Appendix C for details): (1) **Extraction of the correct answer**. Undergraduate medical students first extract the correct answer directly from the original journal's Diagnosis section. (2) **Identification of distractors from source material**. Any misdiagnoses or inappropriate treatment procedures explicitly mentioned in the case discussions are collected as initial distractors. This step typically results in up to five distractors. (3) **Augmentation with LLM-generated candidates**. The constructed question stem is then submitted to GPT-4o, which is instructed to generate the five most plausible and contextually relevant diagnoses, ensuring that each option is clear, precise, and medically valid. (4) **Merging and refinement**. GPT-4o subsequently merges the options from Steps 2 and 3, removing ambiguous or duplicate items, to produce a final, concise set containing one correct answer alongside multiple high-quality distractors.

**Specialty Categorization.** Finally, we classify all questions into 12 medical specialties based on the diseases indicated by the correct answers. The detailed categorization is provided in the Appendix C.3.

### 3.3 CONSTRUCTION OF CLINBENCH$_{MM}$

In real-world clinical scenarios, physicians rely not only on patient textual descriptions but also on medical images from clinical examinations to make informed decisions. To better reflect this multimodal scenario of medical diagnosis,

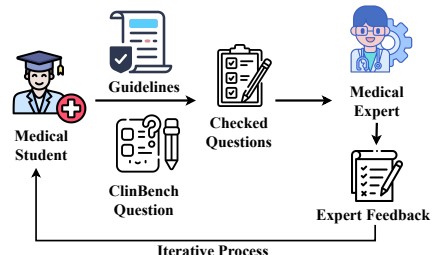

Figure 3: The human check pipeline for Clin-Bench.

we construct a multimodal version of the ClinBench benchmark dataset. Specifically, we employ

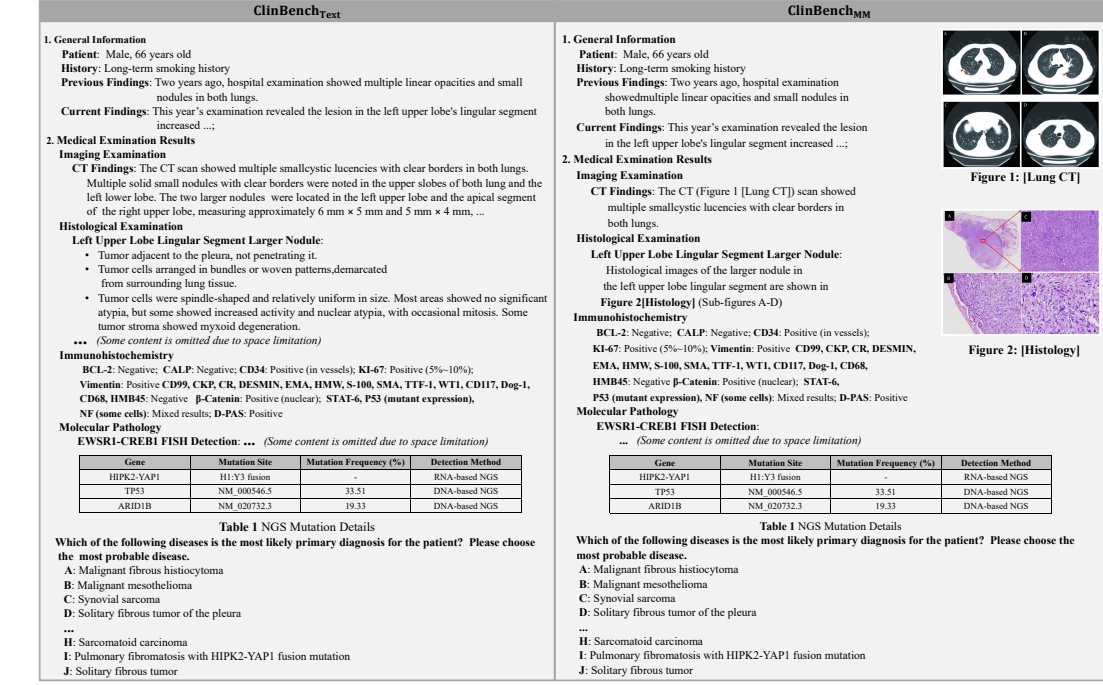

Figure 4: Two demos of ClinBench_Text and ClinBench_MM, respectively. More cases are shown in Appendix F

undergraduate students to curate ClinBench_MM based on questions from ClinBench_Text by removing textual content that describes the associated medical images.

To ensure annotation consistency and quality, we provide the annotators with detailed guidelines (see Appendix C) that instruct them on how to identify and remove image-referential text from the question stems. If a question contains a low-quality image or one that lacks clinically relevant information, it is excluded from the dataset. Following this procedure, we obtain over 2,000 multimodal questions, forming the ClinBench_MM benchmark. Compared to ClinBench_Text, the question stems in ClinBench_MM exclude descriptions related to medical images. ClinBench_MM challenges models to accurately extract and reason over visual content from medical images, providing a rigorous assessment of their multimodal understanding and diagnostic capabilities.

### 3.4 HUMAN EXPERT CHECK

We summarize the potential issues that may arise during the question construction process:

(1) **Incorrect Question Stem:** The constructed question stem may **omit critical medical information** present in the original case report. Additionally, when constructing ClinBench_MM questions, students are required to manually remove textual descriptions of image-related content from the stem. This process may further result in the inadvertent loss of important information essential for accurately understanding or answering the question.

(2) **Inappropriate Candidate Options:** During the option merging process, some candidate options may **overlap with the correct answer** or be ambiguously phrased, making it difficult to ensure a clear and unambiguous set of choices.

**Two-level Human Check.** To ensure the quality of the ClinBench benchmark, we implement a rigorous two-level human check process, and the process is shown in Figure 3. First, medical undergraduate students review the question stems and options to ensure clarity and accuracy. Then, we invite experienced practicing physicians (e.g., attending doctors) to conduct a sampling-based inspection. If any quality issues are identified, they are systematically summarized and fed back to

medical students for targeted revision. This iterative process maintains the overall reliability and quality of the benchmark. More detailed process is shown in Appendix C.

| Model | GH | Surg | Neuro | Oph/ENT | DI | Resp | Dent | OG | Ortho | Cardio | Ped | Avg |
|---|---|---|---|---|---|---|---|---|---|---|---|---|
| *General Large Language Models* | | | | | | | | | | | | |
| GPT-4o-2024-11-20 | 35.7 | 42.4 | 37.3 | 29.8 | 34.0 | 40.6 | 38.3 | 31.7 | 35.4 | 39.8 | 39.0 | 37.4 |
| Deepseek-V3 | **44.9** | **45.5** | 45.2 | **39.7** | 40.9 | 42.4 | **55.0** | 36.6 | 43.4 | 46.4 | 44.1 | **44.3** |
| Grok-3 | 38.5 | 38.4 | 40.8 | 33.1 | 35.5 | 35.9 | 40.0 | 42.7 | 48.5 | 40.4 | 28.8 | 38.7 |
| Phi-4 | 35.2 | 29.3 | 37.0 | 16.5 | 30.5 | 27.1 | 35.0 | 30.5 | 30.3 | 33.8 | 30.5 | 32.3 |
| Llama-3.1-8B-Instruct | 20.8 | 23.2 | 27.4 | 19.8 | 23.7 | 18.8 | 23.3 | 25.6 | 16.2 | 27.3 | 33.9 | 23.7 |
| Llama-3.1-70B-Instruct | 30.0 | 29.3 | 34.3 | 29.8 | 30.1 | 28.2 | 33.3 | 28.1 | 29.3 | 32.0 | 28.8 | 30.9 |
| Qwen-2.5-32B-Instruct | 27.1 | 24.2 | 31.9 | 22.3 | 18.7 | 25.9 | 23.3 | 23.2 | 27.3 | 29.0 | 32.2 | 26.9 |
| Gemma-3-12B-it | 12.4 | 25.3 | 21.9 | 18.2 | 16.3 | 15.9 | 25.0 | 23.2 | 22.0 | 19.2 | 15.3 | 17.4 |
| Claude-Opus-4-20250514 | 42.4 | 41.4 | **46.6** | 32.2 | **45.8** | **43.5** | 33.3 | 40.2 | 50.5 | 47.3 | 45.8 | **44.3** |
| Qwen3-32B | 30.8 | 37.3 | 33.6 | 28.1 | 35.5 | 35.6 | 28.9 | 36.7 | 32.9 | 34.3 | 35.0 | 35.6 |
| Qwen3-8B | 25.8 | 28.3 | 28.8 | 27.2 | 22.1 | 21.2 | 25.0 | 22.0 | 22.2 | 29.0 | 18.6 | 25.7 |
| *General Large Reasoning Models* | | | | | | | | | | | | |
| GPT-5-2025-08-07 | 52.1 | 53.5 | **57.9** | 44.6 | 50.7 | 45.3 | 41.7 | 51.2 | **50.5** | 52.0 | 47.5 | 51.5 |
| Gemini-2.5-pro | **56.3** | 49.5 | 54.1 | **47.9** | **54.7** | 47.1 | 43.3 | 53.7 | 46.7 | 53.3 | 49.2 | **52.6** |
| Deepseek-R1 | 47.4 | 46.5 | 46.9 | 40.5 | 40.9 | 41.8 | **50.0** | 34.2 | 43.4 | 47.0 | 42.4 | 44.8 |
| o3-mini-2025-01-31 | 36.6 | 37.4 | 44.5 | 33.1 | 37.4 | 35.9 | 31.7 | 40.4 | 39.5 | 39.0 | 38.8 | 38.8 |
| o4-mini-2025-04-16 | 45.2 | 46.5 | 51.4 | 41.3 | 41.9 | 42.9 | 38.3 | 46.5 | 45.8 | 40.7 | 45.4 | 45.4 |
| OpenAI-o1-mini | 36.2 | 33.3 | 34.9 | 25.6 | 30.1 | 31.2 | 30.0 | 35.4 | 32.3 | 34.1 | 33.9 | 33.5 |
| Qwen-3-235b-a22b | 40.2 | 31.3 | 44.2 | 30.6 | 38.9 | 36.5 | 43.3 | 31.7 | 34.3 | 27.3 | 44.1 | 36.7 |
| Llama-4-maverick | 46.2 | 40.4 | 46.9 | 36.4 | 48.3 | 40.6 | 40.0 | 39.0 | 44.4 | 44.6 | 42.4 | 44.4 |
| R1-Distill-Qwen-32B | 34.0 | 36.4 | 38.4 | 28.1 | 36.0 | 31.2 | 26.7 | 29.3 | 26.3 | 33.5 | 35.6 | 33.9 |
| *Medical Large Language Models* | | | | | | | | | | | | |
| Llama-3-8B-UltraMedical | 13.4 | 18.3 | 19.0 | 14.9 | 19.8 | 15.9 | 15.0 | 19.5 | 14.1 | 19.2 | 17.8 | 17.1 |
| MedGemma-27B-it | 19.1 | 36.7 | 31.2 | 35.8 | 17.9 | 16.5 | 18.0 | 18.1 | 21.9 | 11.7 | 18.8 | 19.1 |
| Llama-3-70B-UltraMedical | 28.5 | 33.0 | 34.2 | 31.7 | 33.6 | 31.8 | 36.7 | 34.2 | 31.3 | 32.0 | 31.4 | 31.1 |
| Llama-3-OpenBioLLM-8B | 17.1 | 23.2 | 24.7 | 19.8 | 23.7 | 20.6 | 26.7 | 23.2 | 20.2 | 23.7 | 23.7 | 22.8 |
| Llama-3-OpenBioLLM-70B | 38.0 | 33.3 | 37.0 | 33.1 | 32.0 | 38.8 | 36.7 | 40.2 | 37.4 | 38.9 | 39.0 | 37.2 |
| Baichuan-M2-32B | **39.9** | 30.3 | 37.7 | 32.2 | 31.0 | 30.6 | 31.7 | 36.6 | 28.3 | 42.0 | 38.9 | 36.5 |
| HuatuoGPT-o1-7B | 24.3 | 25.3 | 27.7 | 21.5 | 26.6 | 21.2 | 26.7 | 23.2 | 26.3 | 34.4 | 20.3 | 26.5 |
| HuatuoGPT-o1-70B | 39.0 | 43.4 | 42.8 | 39.7 | 38.9 | 42.4 | 46.7 | 39.0 | 40.4 | 43.7 | 42.4 | 39.2 |
| Human Experts | 58.0 | 28.0 | 24.0 | 22.0 | - | 58.0 | 12.0 | 52.0 | 16.0 | 60.0 | 54.0 | 38.4 |

Table 3: Performance (accuracy) of various models across medical specialties on ClinBench$_{\text{Text}}$. **Bold** indicates the best performance, and underlined indicates the second best. Specialty abbreviations: GH (Gastroenterology and Hematology), Surg (Surgery), Neuro (Neurosciences), Oph/ENT (Ophthalmology and ENT), DI (Dermatology and Immunology), Resp (Respiratory and Thoracic Medicine), Dent (Dentistry), OG (Obstetrics and Gynecology), Ortho (Orthopedics), Cardio (Cardiovascular and Internal Medicine), Ped (Pediatrics). Green indicates questions assessed by human experts within their own specialty, while red denotes questions out of their specialty.

## 4 EXPERIMENTS

### 4.1 EXPERIMENTAL SETUP

We evaluate all models under a zero-shot prompt setting. Models with fewer than 32B parameters are evaluated locally using 8 * A800 GPUs. During evaluation, we set the temperature to $t = 0.6$ and report the average results over three independent runs. For models larger than 32B parameters, we use the official APIs for evaluation. Detailed evaluation prompts are provided in Appendix E.

**Evaluated Models** We conduct evaluations on a wide range of LLMs and large multimodal models (LMMs) using ClinBench$_{\text{Text}}$ and ClinBench$_{\text{MM}}$, respectively. Our benchmark includes both *proprietary* and *open-source models*, and additionally covers advanced large *reasoning* medical models, with a focus on capturing the latest advancements in medical reasoning capabilities. Detailed information of models is shown in Appendix E.1.

### 4.2 MAIN RESULTS

Tables 3 and 4 present the main results on ClinBench$_{\text{Text}}$ and ClinBench$_{\text{MM}}$, respectively.

| Model | GH | Surg | Neuro | Oph/ENT | DI | Resp | Dent | OG | Ortho | Cardio | Ped | Avg |
|---|---|---|---|---|---|---|---|---|---|---|---|---|
| Doubao-1.5-Vision-Pro-32k | 29.2 | 34.0 | 34.0 | 33.3 | 33.7 | 28.2 | 20.7 | 34.2 | 33.7 | 36.0 | 28.8 | 32.4 |
| GPT-4o | **35.7** | **37.1** | 38.9 | 32.5 | **39.6** | **38.8** | **37.9** | **41.5** | **41.8** | 37.2 | 33.9 | **37.8** |
| Gemini-2.0-Flash | 27.7 | 34.0 | 39.6 | 28.3 | 28.7 | 31.2 | 27.6 | 31.7 | 29.6 | 33.5 | 28.8 | 31.6 |
| Gemma-3-27B-It | 22.7 | 27.8 | 28.5 | 20.8 | 24.8 | 24.7 | 19.0 | 25.6 | 29.6 | 26.6 | 27.1 | 25.3 |
| Internvl3-14B-It | 18.0 | 33.3 | **50.0** | 28.6 | 20.0 | 23.1 | 20.0 | 30.0 | 26.7 | **41.9** | **40.0** | 30.2 |
| Llama-3.2-11B-Vision-It | 20.8 | 33.3 | 25.8 | 21.4 | 24.0 | 22.4 | 26.3 | 31.8 | 12.2 | 21.0 | 20.0 | 23.3 |
| Qwen2.5-VL-72B-It | 24.6 | 35.3 | 29.7 | 19.2 | 29.0 | 25.5 | 18.2 | 27.3 | 34.8 | 30.9 | 22.2 | 27.8 |
| MedGemma-27B-It | 24.7 | 23.7 | 30.6 | **34.8** | 23.6 | 25.3 | 20.0 | 25.0 | 34.2 | 32.2 | 36.8 | 27.6 |
| HuatuoGPT-Vision-7B | 18.6 | 33.0 | 24.7 | 20.8 | 17.7 | 18.5 | 17.5 | 17.5 | 28.6 | 20.3 | 16.9 | 21.0 |
| Human Experts | 60.0 | 30.0 | 28.0 | 22.0 | - | 64.0 | 12.0 | 54.0 | 16.0 | 60.0 | 54.0 | **40.0** |

Table 4: Performance of various models across medical specialties on ClinBench$_{MM}$. **Bold** indicates the best performance, and underlined indicates the second best.

> ***Key Observation 1:*** Large reasoning models consistently outperform non-reasoning counterparts, underscoring the value of test-time scaling in complex clinical tasks.

**Comparison of LLMs.** (1) Overall, even the most advanced LLMs, such as Deepseek-R1 and OpenAI-o4-mini, achieve no more than 45% accuracy, indicating relatively poor performance on ClinBench. This underscores the benchmark's difficulty and its challenge to state-of-the-art LLMs. (2) Generally, large reasoning models outperform non-reasoning counterparts. For instance, Deepseek-R1 and R1-Distill-Qwen-32B show measurable gains over Deepseek-V3 and Qwen2.5-32B-Instruct, highlighting the benefits of test-time scaling in complex clinical contexts. (3) Medical LLMs with comparable parameter sizes do not significantly surpass general LLMs, which may be attributed to: outdated base models (e.g., Llama3, Qwen2), limited alignment with real-world clinical scenarios, and reduced robustness stemming from fine-tuning, potentially impairing performance on tasks beyond standardized medical examinations.

> ***Key Observation 2:*** Medical LLMs of similar size have no clear advantage over general models, probably due to the fact that they rely on outdated base architectures, focus on exam-oriented training rather than diverse real-world clinical tasks.

**Comparison of LMMs.** (1) Consistent with the results observed on ClinBench$_{Text}$, current LMMs demonstrate relatively low performance on ClinBench$_{MM}$. (2) On the other hand, LMMs achieve approximately 30 points (which is close to the average performance on ClinBench$_{Text}$.) even though the key information of diagnostic images is removed in the question stem. This indicates that LMMs are able to effectively utilize medical image information, which contributes to the resolution of complex clinical problems.

## 4.3 LLM v.s. Human Experts

In this section, we aim to assess whether the capabilities of LLMs in complex medical scenarios, such as diagnostic reasoning, have reached the level of human experts. To this end, we compare the performance of LLMs with that of human experts on the ClinBench$_{MM}$ dataset.

**Setting.** We invite experienced physicians from 5 different medical specialties, all of whom hold at least attending-level positions and have extensive clinical diagnostic experience. The detailed information is shown in the Appendix D. For the evaluation, we randomly select 50 questions from each medical specialty, totaling 500 questions, and assign each expert questions from two specialties: **one within their own area of expertise and another outside of their specialization.** During the answering process, experts are permitted to consult relevant medical literature and textbooks; however, the use of AI-assisted tools is strictly prohibited.

**Medical Insights.** From Table 3, we have the following observations: Human experts can outperform the strongest current LLMs, Deepseek-R1 and OpenAI-o3, in the vast majority of their own specialized fields. On the other hand, experts perform poorly on questions outside their areas of expertise, whereas LLMs demonstrate relatively stable performance across all medical specialties, highlighting their stronger generalization capabilities.

## 4.4 RARE DISEASE TRACK

Rare diseases have long posed significant challenges to the medical community Schieppati et al. (2008); Stoller (2018), primarily due to limited clinical data, insufficient diagnostic knowledge, and a lack of effective treatments. In this work, we include a dedicated rare-disease subset within our dataset. Medical students carefully select 79 rare-disease cases from various medical specialties, strictly adhering to internationally recognized rare-disease catalogs[3]. This rare disease subset provides a specialized evaluation track for LLMs, which is beneficial for advancing LLMs to overcome the challenges of rare diseases in human medicine.

As shown in Table 5, we observe that the performance of LLMs on rare diseases is significantly lower than on non-rare diseases, highlighting the challenge that rare diseases pose to current LLMs. Additionally, medical LLMs such as Llama-3-8B-UltraMedical and HuatuoGPT-o1-7B exhibit relatively smaller performance gaps between rare and non-rare diseases. This may be attributed to their training on more medical texts, including materials related to rare diseases, enabling them to achieve better diagnostic capabilities in rare diseases.

| Model | Rare Acc | Non-Rare Acc |
|---|---|---|
| GPT-4o | 23.1 $_{-16.0}$ | 39.1 |
| Deepseek-R1 | 30.3 $_{-14.9}$ | 45.2 |
| Qwen2.5-7B-Instruct | 8.1 $_{-12.2}$ | 20.3 |
| Llama-3-8B-UltraMedical | 12.5 $_{-11.2}$ | 23.7 |
| HuatuoGPT-o1-7B | 19.5 $_{-7.7}$ | 27.2 |

Table 5: Comparison of model accuracy (%) for rare and non-rare diseases on ClinBench.

## 4.5 RELIABILITY

In clinical settings, particularly for diagnosis, the consequences of a single erroneous model response can outweigh the benefits of numerous correct ones. To rigorously quantify model reliability under such demanding conditions, we evaluate the *worst@k* performance on the Clin-Bench benchmark. We define this metric as the minimum score achieved across $k$ independent model responses. A steeper decline in the *worst@k* score as $k$ increases signifies greater performance volatility and, consequently, lower model reliability.

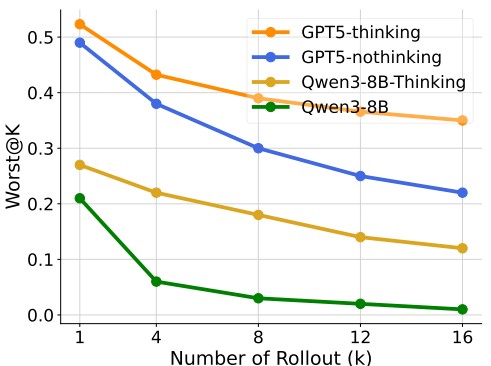

Figure 5: Plot of worst-case ClinBench scores for GPT5 and Qwen3-8B models across different rollouts (k).

We investigate this by testing both a closed-source model (GPT-5 OpenAI (2024)) and an open-source model (Qwen3-8B Yang et al. (2025)), comparing their performance with and without reasoning-augmented prompting. As depicted in Figure 5, our analysis of *worst@16* performance reveals two key findings. First, for all values of $k$ from 1 to 16, the reasoning-augmented models consistently outperform their non-reasoning models. Second, and more importantly, the reasoning models exhibit significantly greater stability, as evidenced by a much gentler degradation in their *worst@k* scores. This suggests that incorporating reasoning not only improves overall accuracy but also substantially enhances the model's robustness and trustworthiness for clinical applications.

---

[3] https://www.who.int/standards/classifications/frequently-asked-questions/rare-diseases

> ***Key Observation 3:*** Large reasoning models consistently improve overall accuracy and enhance the model's robustness and trustworthiness for clinical applications.

### 4.6 ERROR ANALYSIS

To identify LLMs' failure modes on ClinBench, we categorized errors into four types (Format Errors, Clinical Context Misunderstanding, Medical Knowledge Deficiency, Clinical Misdiagnosis) and analyzed 100 random error cases per model in Appendix E.3. The results revealed distinct error patterns: HuatuoGPT-o1-7B had the highest Clinical Misdiagnosis rate (54%), while o3-mini-2025-01-31 showed the highest Medical Knowledge Deficiency rate (51%), highlighting varying weaknesses of LLMs in complex clinical reasoning.

## 5 RELATED WORK

**Medical LLMs.** The success of LLMs has sparked interest in creating medical-specific models, leading to the emergence of numerous powerful medical LLMs (Nori et al., 2023; Saab et al., 2024; Li et al., 2024). For example, UltraMedical collections (Zhang et al., 2024) refine LlaMA-3 models with premium datasets, achieving top benchmark performance and advancing online preference learning. Baichuan-M2 Dou et al. (2025) enhanced the capability of language models in medical diagnosis tasks through a large-scale and realistic dynamic validation system. BioMistral (Labrak et al., 2024), an open-source model pre-trained on PubMed Central, excels in English medical QA tasks. HuatuoGPT series (Chen et al., 2024a;b; Zhang et al., 2023a), trained on high-quality medical data, significantly enhances automated capabilities in diagnosis, triage, and medical imaging, providing valuable support for clinical decision-making and patient care. Building on this trend, recent models Yan et al. (2025); Chen et al. (2025) such as Lingshu Xu et al. (2025) and HealthGPT Lin et al. (2025) further advance the field by improving medical reasoning, multimodal understanding, and have demonstrated strong empirical performance across a range of medical benchmarks.

**Medical Benchmarks.** With the advancement of medical LLMs, corresponding benchmarks have also evolved. Early datasets such as MedQA (Jin et al., 2021) and MedMCQA (Pal et al., 2022) focus on multiple-choice questions from USMLE and Indian medical exams, assessing models' factual knowledge across various medical domains. PubMedQA (Jin et al., 2019) emphasizes reasoning over biomedical research abstracts. On the other hand, the emergence of MMLU-Pro (Wang et al., 2024) and GPQA (Frantar et al., 2022) benchmarks provides a more effective evaluation of long-chain reasoning models like OpenAI-o1. Additionally, MedXpert-QA Zuo et al. (2025) introduces expert-level questions derived from advanced medical exams, significantly increasing benchmark difficulty. However, these benchmarks remain predominantly exam-oriented and fail to capture the complexity of real-world clinical scenarios. In this paper, we focus on realistic medical scenarios by constructing a benchmark based on real-world clinical case questions. The recent emergence of HealthBench Arora et al. (2025) has introduced rubric-based evaluation for open-ended medical tasks, where large language models are required to score responses according to predefined criteria. While such a rubric benchmark is suitable for open-ended assessments, it adds additional complexity to the evaluation process. By contrast, our ClinBench adopts a multiple-choice format, enabling more straightforward, consistent, and reliable evaluation.

## 6 CONCLUSION

In this paper, we introduce ClinBench, a medical benchmark specifically designed to simulate real clinical scenarios. This challenging benchmark originates from authoritative medical cases and incorporates detailed patient information and clinical findings, offering a more realistic assessment of LLMs' medical reasoning. Our comparative analysis of over 20 LLMs against medical experts demonstrates the continued strength of human specialists within their domains, while also highlighting the impressive ability of LLMs to generalize across a wider range of medical knowledge, suggesting their potential to complement and enhance clinical expertise.

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

## A  DECLARATION OF LARGE LANGUAGE MODELS (LLMs) USAGE

In the development of this work, LLMs were utilized solely as auxiliary tools, with no involvement in the core ideation and conceptualization of the research—all key research ideas, experimental designs targeting the **ClinBench** medical benchmark, and overall framework of the study were independently formulated by the human authors.

Specifically, the LLM was employed in three limited, non-core capacities: (1) Assisting in the structuring and organization of experimental data related to **ClinBench**, including formatting raw medical data for consistency and readability without altering data integrity or content; (2) Supporting the preliminary collation of experimental result evaluations (e.g., organizing metric values such as perplexity and ROUGE-L into clear, standardized summaries, while all analytical conclusions and interpretations of results remained human-derived); (3) Providing language polishing for the manuscript, such as refining sentence structure, improving expression clarity, and ensuring grammatical accuracy, without modifying the scientific meaning, arguments, or conclusions presented in the text.

The authors confirm full responsibility for all content of this work, including the accuracy of data, validity of experimental analyses, and integrity of scientific claims. No part of the work involves LLM-generated content that could be construed as plagiarism, misrepresentation, or scientific misconduct. Consistent with academic norms, LLMs are not considered contributors or eligible for authorship of this manuscript.

## B  ETHICS STATEMENT

All data used in this work were obtained exclusively from **freely and publicly accessible** sources. We have carefully curated the dataset by retaining only a small and representative subset of the original data. To ensure compliance with **U.S. fair use laws**, all questions underwent rephrasing, and answer options were shuffled to prevent any direct replication. Importantly, the dataset **does not contain any personal, sensitive, or identifiable information**, strictly avoiding any privacy violations or ethical concerns related to personal data. No content involving individual identities, medical records, or confidential information has been included. To mitigate the potential data leakage risks, we refrain from releasing the data sources and request that you **do not share any example of benchmark online**, whether in plain text, image, or any other format.

## C  DETAILED PROCESS FOR CLINBENCH CONSTRUCTION

In this section, we provide a detailed description of the construction process of the **ClinBench** benchmark.

### C.1  DATASET CONSTRUCTION

We recruited 30 medical undergraduate students to assist in the dataset construction process. These students, all majoring in medicine, possess solid medical foundations. We provided them with comprehensive annotation guidelines, instructing them to carefully construct and verify question–answer pairs. Specifically, the guidelines detailed two core tasks: (1) constructing the ClinBench$_{Text}$ questions (see Table 10); and (2) constructing the multimodal ClinBench$_{MM}$ questions (see Table 11). Following these guidelines rigorously, the students successfully constructed a total of 2,014 high-quality questions.

### C.2  HUMAN VERIFICATION

Our human verification process involves two critical steps. First, medical undergraduate students carefully review the question stems and candidate options to ensure clarity and medical accuracy, following detailed guidelines as illustrated in Table 12. Subsequently, experienced practicing physicians (e.g., attending doctors) perform a sampling-based inspection of the reviewed questions. Any identified quality issues are systematically summarized and communicated back to the medical stu-

dents for targeted revision, guided by the criteria provided in Table 13. This iterative feedback and revision mechanism ensures the overall reliability and high quality of the benchmark.

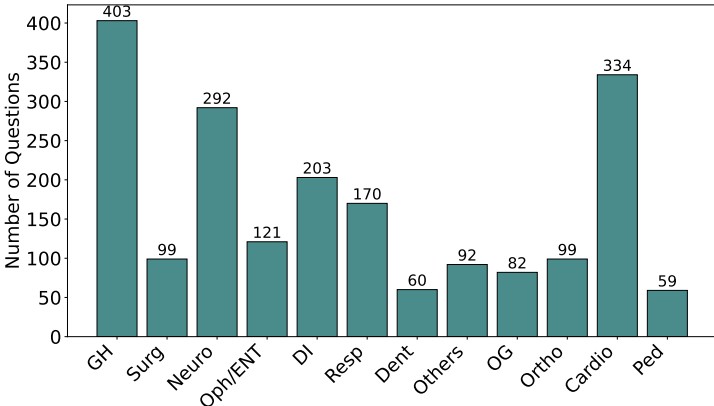

Figure 6: The pipeline for constructing ClinBench. Specialty abbreviations: GH (Gastroenterology and Hematology), Surg (Surgery), Neuro (Neurosciences), Oph/ENT (Ophthalmology and ENT), DI (Dermatology and Immunology), Resp (Respiratory and Thoracic Medicine), Dent (Dentistry), OG (Obstetrics and Gynecology), Ortho (Orthopedics), Cardio (Cardiovascular and Internal Medicine), Ped (Pediatrics).

### C.3 DATA SPECIALTY DISTRIBUTION

We analyze the distribution of medical specialties covered by the ClinBench$_{Text}$ and ClinBench$_{MM}$ datasets, as illustrated in Figure 6. As shown, each medical specialty includes at least 50 questions, ensuring sufficient coverage for comprehensive evaluation.

| Model | GPU Hours (h) | API Quotas ($) |
|---|---|---|
| Deepseek-V3 | - | 2.090 |
| Llama-4-maverick | - | 1.098 |
| o3-mini-2025-01-31 | - | 5.460 |
| HuatuoGPT-o1-7B | 0.42 | - |
| Qwen2.5-7B-Instruct | 0.167 | - |

Table 6: Model Evaluation Costs (All Questions) on ClinBench_Text

### C.4 COMPUTATIONAL COST FOR DATASET CONSTRUCTION AND MODEL EVALUATION

To facilitate reproducibility and transparency of our experiments, we report the detailed computational costs for both dataset construction and model evaluation. All API-based costs were incurred via official service channels, ensuring compliance and consistency. GPU hours refer to actual runtime used for local inference. The evaluation costs for ClinBench_MM are nearly identical to those for ClinBench_Text.

## D EXPERT EVALUATION DETAILS

Our experts are experienced attending physicians from Longgang People's Hospital in Shenzhen, Guangdong Province, China, specializing in Gastroenterology and Hematology, Respiratory and Thoracic Medicine, Obstetrics and Gynecology, Cardiovascular and Internal Medicine, and Pediatrics. For the evaluation, we randomly select 50 questions from each of these

| Dataset Component | API Cost ($) |
|---|---|
| ClinBench (Total) | 720 |
| ClinBench_Text | 620 |
| ClinBench_MM | 100 |

Table 7: Dataset Construction Costs

medical specialties, totaling 500 questions, and assign each expert questions from two specialties: **one within their own area of expertise and another outside of their specialization.** During the answering process, experts are permitted to consult relevant medical literature and textbooks; however, the use of AI-assisted tools is strictly prohibited.

For the evaluation, we developed an online assessment platform based on the streamlit framework, allowing physicians to answer questions following the provided guidelines. A screenshot of the evaluation platform is shown in Figure 9.

# E EXPERIMENTAL DETAILS AND ADDITIONAL RESULTS

## E.1 MODELS

**Large Language Models:** We evaluate a wide range of LLMs on ClinBench$_{\text{Text}}$. The general LLMs include GPT-4o-2024-11-20 OpenAI (2024), DeepSeek-V3 Liu et al. (2024), Grok-3, Phi-4 Abdin et al. (2024), LLaMA3.1-70B-Instruct, LLaMA3.1-8B-Instruct, Qwen2.5-32B-Instruct, Qwen2.5-7B-Instruct and Qwen3-8B Yang et al. (2025). Gemma-3-12B-it Team et al. (2025) We also include general reasoning models such as DeepSeek-R1, DeepSeek-R1-Distill-Qwen-32B Guo et al. (2025), Qwen-3-235b-a22b Yang et al. (2025), OpenAI-o1, OpenAI-o3-mini, OpenAI-o3-mini [4], Claude-opus-4-20250514 Anthropic, Grok-3, GPT-5 OpenAI (2024), Llama-4-maverick[5]. In the medical domain, we assess medical-domain LLMs including OpenBioLLM-8B Pal & Sankarasubbu (2024), and UltraMedical-8B Zhang et al. (2024),Baichuan-M2-32B Dou et al. (2025), as well as the medical reasoning model HuatuoGPT-o1 Chen et al. (2024a).

**Large Multimodal Models (LMMs):** We evaluate several large multimodal models on ClinBench$_{\text{MM}}$, including proprietary models such as GPT-4o-2024-11-20 OpenAI (2024), Gemini-2.0-Flash-001 Google (2024), as well as open-source models such as Doubao-1.5-Vision-Pro-32k, Internvl3-14B Zhu et al. (2025), Qwen2.5-VL-72B Team (2025)and Llama-3.2-11B-Vision Grattafiori et al. (2024)

We evaluate all models under a zero-shot prompt setting. For models such as llama-4-maverick, GPT-4o, Deepseek-V3, Gemini-2.5-Pro-Exp-03-25, Deepseek-R1, Grok-3, doubao-1-5-pro-32k-250115, o1-mini, o3-mini, o4-mini, and Qwen3-235B-a22b, we utilize the official APIs provided by the official. The remaining models are evaluated on a local setup consisting of 8 A800 80GB GPUs. During evaluation, we set the temperature to $t = 0.6$ and report the average results over three independent runs. The prompts used are illustrated in Figure 7 and Figure 8.

## E.2 DATA LEAKAGE ANALYSIS

To evaluate the potential risk of data leakage in the **ClinBench** benchmark, we follow work Xu et al. (2024) by employing perplexity (PPL) and N-gram-based metrics (ROUGE-L and edit distance similarity) as evaluation criteria. Specifically, we concatenate the original question with a prompt such as "Answer:" as input and calculate the model's perplexity on the generated output. Additionally, to assess the similarity between the model's generated rationale and the reference explanations we collected, we compute both ROUGE-L scores and edit distance similarity. We hypothesize that if "data leakage" occurred, the models would have previously learned specific reference texts, resulting in lower perplexity (PPL), higher textual overlap (Rouge-L), and higher edit distance similarity (EDS). Conversely, if no leakage occurred, we would expect the opposite pattern.

We evaluate three models, including GPT-4o, LLaMA-3.1-70B-Instruct, and Qwen2.5-72B-Instruct, on ClinBench. As shown in Table 8:

- **PPL (Perplexity):** All models exhibit extremely high PPL values (e.g., GPT-4o: $1.18 \times 10^{120}$, LLaMA-3.1-70B-Instruct: $9.73 \times 10^{146}$), indicating substantial difficulty in predicting the reference texts and lack of prior exposure.
- **Rouge-L (Textual Overlap):** Rouge-L scores are uniformly low (maximum 0.1794), reflecting minimal textual overlap with reference texts and no evidence of memorization.

---

[4] https://openai.com/o1/
[5] https://ai.meta.com/blog/llama-4-multimodal-intelligence/

| Models | PPL ↑ | Rouge-L ↓ | EDS ↓ |
|---|---|---|---|
| GPT-4o | 1.18E+120 | 0.1712 | 0.2391 |
| Qwen2.5-72B-Instruct | 1.12E+115 | 0.1794 | 0.2493 |
| LLaMA-3.1-70B-Instruct | 9.73E+146 | 0.1597 | 0.2285 |

Table 8: Data leakage analysis results on different models. PPL denotes Perplexity, and EDS stands for Edit Distance Similarity.

- **EDS (Edit Distance Similarity):** EDS values are similarly low (maximum 0.2493), consistent with minimal content overlap and absence of memorization.

Our analysis finds no evidence of data leakage. This is primarily due to: (1) the questions in ClinBench being sourced from professional medical case reports absent from the models' training corpora; and (2) the inherent complexity and rich clinical context making memorization or direct reproduction from prior exposure highly unlikely. These findings collectively contradict the data leakage hypothesis, confirming that ClinBench poses minimal risk of leakage and supports its validity and robustness as a medical benchmark.

### E.3 ERROR ANALYSIS

To better understand common failure modes of LLMs on ClinBench, we categorized errors into the following four types, each defined as follows:

1. **Format Errors:** The model output does not conform to the required answer format or provided instructions (e.g., missing required fields, incorrect answer structure, or extraneous text).

2. **Clinical Context Misunderstanding:** The response reflects an incorrect interpretation of the clinical scenario or question context, leading to inappropriate reasoning paths.

3. **Medical Knowledge Deficiency:** The model correctly interprets the question but produces an incorrect answer due to insufficient or inaccurate medical knowledge.

4. **Clinical Misdiagnosis:** The model applies correct medical knowledge and general reasoning but ultimately arrives at the wrong clinical conclusion or diagnosis.

We conducted a detailed statistical analysis of these error types across multiple models, with each model evaluated on a random sample of 100 error cases. The results are summarized in Table E.3.

| Model | Format Errors | Clinical Context Misunderstanding | Medical Knowledge Deficiency | Clinical Misdiagnosis |
|---|---|---|---|---|
| Deepseek-V3 | 10 | 30 | 39 | 21 |
| HuatuoGPT-o1-7B | 10 | 21 | 15 | 54 |
| Llama-3.1-8B-Instruct | 11 | 29 | 30 | 40 |
| o3-mini-2025-01-31 | 12 | 22 | 51 | 15 |
| Qwen2.5-7B-Instruct | 11 | 21 | 34 | 34 |

Table 9: Error Analysis Across Different Models (%)

These results reveal distinct error distribution patterns across models. For example, HuatuoGPT-o1-7B exhibits a notably high rate of *Clinical Misdiagnosis* (54%), whereas o3-mini-2025-01-31 shows the highest incidence of *Medical Knowledge Deficiency* (51%). Such findings highlight the varying weaknesses of LLMs in complex clinical reasoning tasks.

## F CASES OF CLINBENCH

We present two specific cases, one multimodal and one text-based, each including detailed questions, options, and the model's responses (from GPT-4o and Llama-3.2-11B-Vision) in Figure 10 and 11.

---

**Evaluation Prompt for ClinBench_Text**

```
Question:  {Question}

Options:  {Options}
```

**Instruction:** Given the following multiple-choice question and options, provide a concise answer based on accurate knowledge. Conclude your response with the correct option in the format: `The answer is [Option].`

---

Figure 7: Evaluation Prompt for ClinBench_Text

---

**Evaluation Prompt for ClinBench_MM**

```
Question:  {Question}

Images:  {Images}
Options:  {Options}
```

**Instruction:** Given the following multiple-choice question, associated medical images, and options, analyze the images and question to provide a concise, accurate answer based on medical knowledge. Conclude your response with the correct option in the format: `The answer is [Option].`

---

Figure 8: Evaluation Prompt for ClinBench_MM

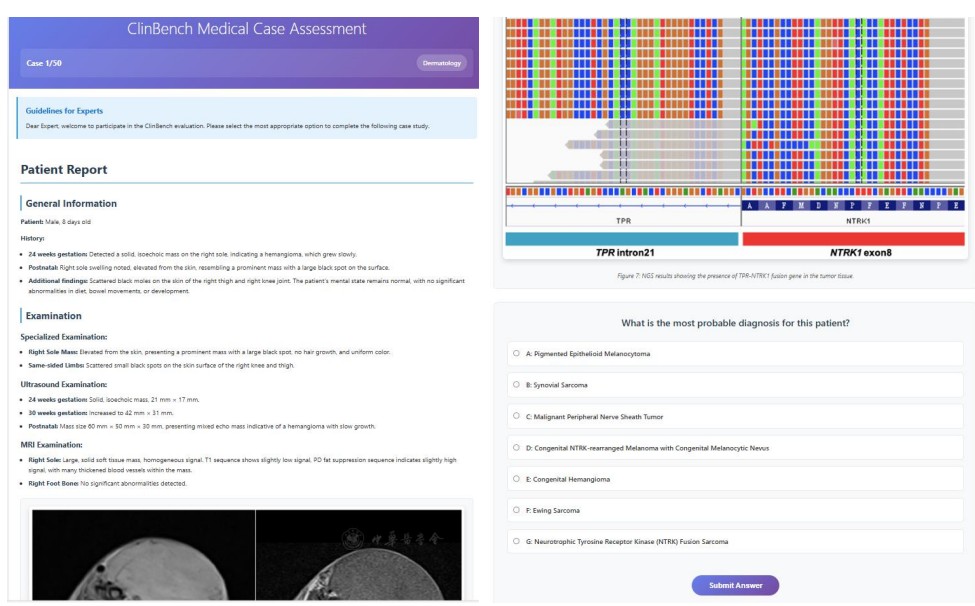

Figure 9: The online evulation of ClinBench.

---

**Guideline for Constructing ClinBenchText**

---

Dear contributor, welcome to the ClinBench question construction task. You are now assigned to build high-quality samples for the ClinBenchText dataset. Please carefully follow the guidelines below.

**1. Constructing the Question Stem**
**Objective:** Use the original content of the medical journal to construct a question stem, while minimizing modification of the source content.
    1.1    For Chinese-language journals, directly extract the *Patient Information* and *Medical Examination Results* sections.
    1.2    For English-language journals, locate the *Patient Information* and *Medical Examination Results* and use a professional translation tool to translate them into fluent, accurate English. Discard journals that do not contain both sections.
    1.3    If the case contains images (e.g., X-ray, MRI, CT), **remove the images**, but ensure the question stem includes a concise and accurate textual description of the image. If such a description cannot be provided, discard the journal.
    1.4    The question stem should be self-contained, medically accurate, and unambiguous, providing enough context for diagnosis.
**2. Constructing Candidate Answer Options**
**Objective:** Create a set of multiple-choice options, including one correct answer and several plausible distractors.
    2.1    **Identifying the Correct Answer:** Extract the final diagnosis from the Diagnosis/Conclusion section. The correct answer must be medically precise and supported by evidence from the journal.
    2.2    **Selecting Distractors from the Discussion:** Extract suspected or differential diagnoses from the Discussion section to serve as medically relevant distractors.
    2.3    **Generating Additional Distractors Using AI:** Use GPT-4o or a similar model to generate plausible but incorrect distractors. They should reflect realistic diagnostic confusion within the relevant medical context.
    2.4    **Merging Options:** Use an AI model or human review to ensure:
    (1)    Exactly one correct answer is included.
    (2)    No ambiguity or semantic overlap among options.
    (3)    At least six options in total.
    2.5    **Final Verification:** All options must be:
    (1)    Medically plausible and realistic.
    (2)    Mutually exclusive and clearly distinct.
    (3)    Professionally formatted, with consistent language and style.
**3. Saving the Output**
Please save each constructed question in a single-entry JSON format, as shown below:

```
{
  "id": "ClinBench-text-00001",
  "question_stem": "",
  "options": "",
  "answer": "",
  }
```

---

Table 10: Guideline for constructing ClinBenchText questions.

**Guideline for Constructing ClinBenchMM**

Dear contributor, welcome to the ClinBench multimodal question construction task. Your task is to build multimodal (text-image) questions based on previously constructed ClinBenchText questions. Please carefully follow the guidelines below.

**1. Adding Medical Images**
**Objective:** Enhance the existing ClinBenchText question stems by appropriately inserting medical images from the original case journals.
    1.1    Identify suitable medical images (e.g., X-ray, MRI, CT scans, histological images) within the original case journal.
    1.2    Insert the identified image at a relevant location within the question stem. Ensure the image directly supports and complements the textual content.
    1.3    If no suitable medical images can be found or inserted into the text, discard the question.
**2. Modifying the Question Stem**
**Objective:** Adjust the existing text-based question stem by removing textual descriptions of medical examination results depicted in the inserted images.
    2.1    Remove explicit textual descriptions of medical findings or examination results that are clearly illustrated by the medical image.
    2.2    Retain only concise image references within the text (e.g., *"The patient's lung condition is shown in Figure 1."*).
    2.3    Ensure the modified question stem remains clear, medically accurate, and contextually complete when combined with the inserted image.
**3. Saving the Output**
Please save each multimodal question as a JSON file following the single-entry format below:

```
{
  "id": "ClinBench-mm-00001",
  "question_stem": "",
  "options": "",
  "answer": "",
  "image": ""
}
```

Ensure each image filename corresponds clearly with the JSON entry and store images in a designated folder. Submit JSON files along with associated images.

Table 11: Guideline for constructing ClinBenchMM multimodal questions.

**Guideline for ClinBench Verification (Medical Students)**

Dear contributor, welcome to the ClinBench question quality check task. Your role is critical to ensuring the high quality of constructed ClinBench$_{\text{Text}}$ and ClinBench$_{\text{MM}}$ questions. Please carefully follow the guidelines below.

**1. Checking for Missing Information**
**Objective:** Identify and correct significant information omissions in the constructed question stem compared to the original medical journal case.
    1.1    Carefully review the constructed question stem against the original medical journal case to identify missing critical medical information necessary for understanding or answering the question.
    1.2    Specifically, for ClinBench$_{\text{MM}}$ questions, pay special attention to information inadvertently omitted during the manual removal of image-related descriptions.
    1.3    If you identify missing crucial information, supplement the question stem by adding the necessary content clearly and concisely, ensuring the revised question stem provides adequate context for accurate diagnosis.
**2. Checking for Inappropriate Candidate Options**
**Objective:** Evaluate and refine the candidate answer options to ensure clarity, distinctiveness, and medical accuracy.
    2.1    Review candidate options carefully for any overlaps with the correct answer or ambiguous phrasing that may confuse test-takers or reduce question clarity.
    2.2    Remove or revise any candidate options identified as overlapping with the correct answer or ambiguously phrased, ensuring the final set of options is clear, distinct, and unambiguous.
**3. Saving the Corrected Output**
After performing the above checks and corrections, save each question in a single-entry JSON format as follows:

```
{
  "id": "ClinBench-check-00001",
  "question_stem": "(Corrected and complete question stem)",
  "options": "(Verified and corrected candidate options)",
  "answer": "(Confirmed correct answer)",
  "image": "(Image filename if applicable, otherwise empty)"
}
```

Ensure each corrected JSON file is named appropriately and clearly linked with any associated images.

Table 12: Guideline for checking and verifying constructed ClinBench questions.

**Guideline for Expert Review of ClinBench.**

Dear expert reviewer, thank you for participating in the quality assurance of the ClinBench dataset. Your expertise is crucial for ensuring the accuracy and clinical relevance of our medical questions. Please carefully follow the guidelines outlined below.

**Objective:** Evaluate each question comprehensively from the following four perspectives: clarity of expression, clinical rationality, factual accuracy, and appropriateness of candidate options.

**Evaluation Criteria:**

- **Clarity of Expression:** Ensure the question stem and options are clearly phrased, understandable, and professionally articulated.

- **Factual Accuracy:** Verify that the question stem and correct answer are medically accurate, evidence-based, and free from factual errors.

- **Appropriateness of Candidate Options:** Check whether the options provided are medically plausible, clearly distinct, non-overlapping, and appropriately challenging.

**Providing Feedback:**

If any issues or concerns are identified during your evaluation, clearly outline your feedback as follows:

- Describe the specific issue identified (clarity, rationality, accuracy, or appropriateness).

- Provide detailed recommendations or corrections for resolving the identified issue.

- Highlight any relevant medical literature or sources that support your feedback if necessary.

Table 13: Guideline for expert review of ClinBench questions.

**F.1 Example of GPT-4o**

**Text-Modal task**

**Question**:
Patient Information
** General Information:
** Patient: Female, 38 years old
** Chief Complaint:Abnormal urine test for 2 months
** Symptoms: None reported (rash, purpura, joint pain, fever, Raynaud's phenomenon, nausea, vomiting)
** Past Medical History:
- Hepatitis B carrier, no antiviral treatment,
- No history of hepatitis C
- Hypertension detected in March 2022, no antihypertensive therapy started
** Surgical History:** Cholecystectomy in March 2022
** Lifestyle:** Non-smoker, non-drinker.
** Family History: No familial hereditary disease or cancer.
** Physical Examination:
- Anemic appearance, no rashes, no enlarged lymph nodes
- Cardiovascular, respiratory, and abdominal examinations: Negative
** Electron Microscopy: - Features consistent with immune complex-mediated glomerulonephritis

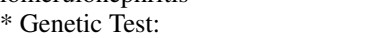
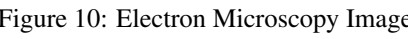
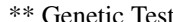

Figure 10: Electron Microscopy Image

** Genetic Test:
** MYD88 L265P Mutation: Negative (result may be influenced by low abnormal cell count)
**Answer Choices:** (A) Chronic Lymphocytic Leukemia (B) Essential Mixed Cryoglobulinemia (C) Heavy Chain Disease (D) Membranoproliferative Glomerulonephritis type I (E) Monoclonal Gammopathy of Renal Significance (F) Monoclonal Gammopathy of Undetermined Significance (G) Type I Cryoglobulinemic Glomerulonephritis (H) Type II Cryoglobulinemic Glomerulonephritis (I) Waldenstrom's Macroglobulinemia
**Correct Answer:** (I)
**Model Answer:** (E)
**Model Analysis:**
To determine the correct answer, the patient presents with abnormal urine findings (proteinuria, hematuria, elevated ACR, and UTP) and has laboratory and biopsy results indicative of immune complex-mediated glomerulonephritis. Key findings include:
** Renal Biopsy: Immune complexes (IgM, C3, Kappa), suggest monoclonal paraprotein involvement.
** Blood and Urine Studies: Presence of monoclonal protein (IgM and Kappa light chains), increased free Kappa, positive SPE (Serum Protein Electrophoresis), and unchanged renal structure.
** Thus, the findings fulfill the criteria for **Monoclonal Gammopathy of Renal Significance (MGRS)**, wherein monoclonal proteins cause renal disease without meeting criteria for malignancy.
** The answer is E.

**F.2 Example of Llama-vision**

## Multi-Modal task

**Question**:
** Clinical Case Report
** Patient Information
** Occupation: Raw material crusher in an industrial ceramics factory
** Case Details
The patient was admitted on October 16, 2018, due to recurrent cough and sputum production for over 10 years, which had worsened with chest tightness for 5 days.
** Initial Examination
** Symptoms: Lip cyanosis, reduced breath sounds in the left lung, bilateral basal lung moist rales; no pathological murmurs in valve auscultation areas.
** Supplementary Check
** Examination Findings: Palpable crepitus in the neck, left supraclavicular fossa, and left chest wall. Auscultation: Diffuse wheezing, numerous moist rales heard at lung bases.
** Chest CT Findings:
- Multiple gas shadows in mediastinum, supraclavicular fossa, and subcutaneous left chest wall.
- Small area of pneumothorax on the left side.
- Nutritional support and appropriate respiration exercises.

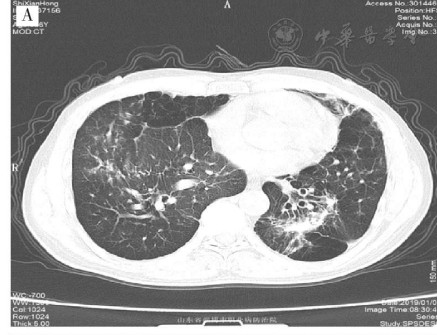
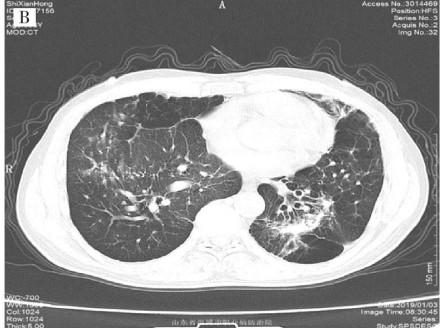

Figure 11: Chest CT Image

**Answer Choices:** (A) Chronic Obstructive Pulmonary Disease (B) Community-Acquired Pneumonia (C) Pneumomediastinum (D) Pneumothorax (E) Primary Spontaneous Pneumothorax (F) Pulmonary Fibrosis (G) Secondary Mediastinal Emphysema (H) Type II Cryoglobulinemic Glomerulonephritis (I) Waldenstrom's Macroglobulinemia

**Correct Answer:** (G)
**Model Answer:** (G)
**Model Analysis:**
** Step 1: Analyze the patient's symptoms and medical history.** The patient is a 56-year-old male with a history of recurrent cough and sputum production for over 10 years, which had worsened with chest tightness for 5 days. He works as a raw material crusher in an industrial ceramics factory.
** Step 2: Analyze the exacerbation symptoms and findings.** Aggravated frequent coughing with yellow purulent sputum, severe chest tightness, orthopnea, left-sided neck swelling and pain - Palpable crepitus in the neck, left supraclavicular fossa, and left chest wall - Diffuse wheezing, numerous moist rales heard at lung bases
** Step 3: Determine the most likely diagnosis based on the information provided.** Given the patient's occupation in an industrial ceramics factory, the presence of pneumothorax, pneumomediastinum, and subcutaneous emphysema, along with the history of recurrent respiratory issues, the most likely diagnosis is related to occupational exposure leading to lung damage.
** Answer: G. Secondary Mediastinal Emphysema.

