# OpenReview forum: "Benchmarking LLMs on Authentic Cases from Medical Journals"
_ICLR.cc/2026/Conference — ICLR 2026 Conference Withdrawn Submission_

### Official Review · Reviewer_2djB · 2025-10-27

**Soundness:** 3
**Presentation:** 3
**Contribution:** 2
**Rating:** 6
**Confidence:** 4

**Summary:**

The authors propose in this paper ClinBench, a new textual and multimodal multi-choice benchmark designed to evaluate LLMs in real-world medical reasoning. Authors expose that existing benchmarks often rely on medical exam questions, which do not capture the complexity of real clinical practice. This benchmark addresses this by using authentic clinical cases from medical journals, including full patient information and test results. The benchmark spans 12 medical specialties and over 2,000 questions, with a special track for rare diseases. All cases are expert-reviewed. The proposed study evaluates a large set of LLMs, both open-source and proprietary, and compares their performance to human medical experts. Results show that while humans outperform LLMs in their own specialties, LLMs achieve stronger overall performance across multiple fields.

**Strengths:**

1) The work proposed in the paper covers data collection, benchmarking, and broad evaluation of existing models.

2) The involvement of medical experts ensures both the human verification of the benchmark and the evaluation process.

3) The two-level human check appears as a robust procedure when building the benchmark.

4) The number of models evaluated is very large and covers both proprietary and open source models.

**Weaknesses:**

1) Data contamination was not mentioned or studied in the paper.

2) The study remains fairly quantitative and lacks some qualitative results, particularly in difficult cases, even if a categorization helps to better understand the errors.

3) I was unable to find the availability of the benchmark, nor its conditions of availability and use (user license).

4) No solution is proposed to improve existing models, nor a detailed analysis of difficult cases.

**Questions:**

I have a few questions/remarks regarding the work:

1) The language of the benchmark should be mentioned in the abstract (English here).

2) What are the rights associated with the data collected, not regarding the patients' information, but the papers themselves? As well as with the data "transformed" to implement the benchmark?

3) How to ensure that there is no data contamination in existing LLMs?

4) Figure 1 is never cited. Figure 3 is cited in the text before Figure 2. Same for Table 3 and Table 2. Figure 6 is a little blurry.

---

### Official Review · Reviewer_bAzq · 2025-10-31

**Soundness:** 2
**Presentation:** 3
**Contribution:** 2
**Rating:** 2
**Confidence:** 4

**Summary:**

This paper proposes a new medical multiple-choice question benchmark, ClinBench, constructed from clinical cases in PubMed Central, covering 12 medical specialties with over 2,000 questions. Based on this benchmark, the authors evaluated the performance of over 20 LLMs and compared it with that of human experts. Experimental results indicate that human experts maintain a significant advantage over state-of-the-art LLMs within their own specialties, but perform markedly worse outside their areas of expertise.

**Strengths:**

1. ClinBench is constructed from medical journal cases, making it closer to real-world clinical scenarios. It provides two modalities (text and multimodal), which can to some extent reflect the clinical diagnostic abilities of current LLMs.
2. The authors conducted a systematic evaluation of over 20 LLMs, and also included human expert performance, providing a more intuitive comparison of LLMs’ capabilities relative to human experts.

**Weaknesses:**

1. The task design and construction of ClinBench lack novelty. The authors claim that “[existing] benchmarks remain predominantly exam-oriented and fail to capture the complexity of real-world clinical scenarios.” However, previous works have already developed MCQA-style medical benchmarks based on medical journal cases [1] and medical reports from top-tier tertiary hospitals [2]. The authors do not provide a comparison with these existing studies.
2. In addition, the evaluation format of ClinBench is overly singular. The authors claim that “ClinBench focuses on diagnostic problems because they represent a core, high-stakes clinical challenge that is amenable to objective, verifiable evaluation.” However, focusing solely on diagnostic problems is insufficient to capture the complexity of real-world clinical applications. Furthermore, the authors state that “ClinBench adopts a multiple-choice format, enabling more straightforward, consistent, and reliable evaluation,” but compared with some existing benchmarks with diverse task formats [3,4], ClinBench lacks task diversity and does not reflect the variety of ways LLMs can apply medical knowledge.
3. Moreover, the solvability of ClinBench questions has not been verified. The paper mentions that manual checks mainly ensure that each question contains the complete information from the original case and that the answer options do not overlap. However, it is not guaranteed that the information provided in the original case ensures that the question is truly solvable. Considering that even experienced experts can only achieve around 60% accuracy in their respective specialties, the solvability of ClinBench questions requires further experimental validation.

[1] Perets O, Shoham O B, Grinberg N, et al. CUPCase: Clinically Uncommon Patient Cases and Diagnoses Dataset. AAAI 2025.

[2] Ouyang Z, Qiu Y, Wang L, et al. CliMedBench: A Large-Scale Chinese Benchmark for Evaluating Medical Large Language Models in Clinical Scenarios. EMNLP 2024.

[3] Wu C, Qiu P, Liu J, et al. Towards evaluating and building versatile large language models for medicine. npj Digital Medicine, 2025.

[4] Zhou Y, Liu X, Yan C, et al. Evaluating LLMs Across Multi-Cognitive Levels: From Medical Knowledge Mastery to Scenario-Based Problem Solving. ICML 2025.

**Questions:**

1. Could you provide an analysis of the solvability of ClinBench questions ? Additionally, why can even experienced physicians only achieve around 60% diagnostic accuracy within their own specialties?
2. I noticed that some questions in the submitted supplementary files contain a mixture of Chinese and English (e.g., No. 16 in the text subset). Could you explain why this occurs, and provide more detailed information on the construction and quality control of ClinBench?

---

### Official Review · Reviewer_h4g3 · 2025-11-01

**Soundness:** 3
**Presentation:** 3
**Contribution:** 2
**Rating:** 4
**Confidence:** 4

**Summary:**

This paper introduces ClinBench, a comprehensive benchmark for evaluating LLMs' medical capabilities, derived from real-world clinical cases in PubMed Central. Constructed through a rigorous human review process, ClinBench spans 12 medical specialties and contains over 2,000 MCQ questions, providing a fine-grained assessment of domain-specific knowledge. Using this benchmark, the authors systematically evaluate more than 20 LLMs and compare their performance against human experts. The experimental results reveal a notable domain-specific advantage for human experts who outperform LLMs within their own specialties. However, outside their expertise, human performance drops significantly, whereas LLMs maintain relatively stable results across different specialties.

**Strengths:**

1.	Compared with traditional medical exam-style questions, ClinBench’s items are directly sourced from authoritative medical journals (PubMed Central), preserving complete patient histories, examination results, and imaging data. This design makes the benchmark much closer to real-world diagnostic workflows and clinical decision-making scenarios.
2.	ClinBench was constructed under quality control, involving a two-level manual review and feedback process (medical students and physicians), which helps improve the overall quality of the benchmark.
3.	Based on ClinBench, the authors conducted a systematic evaluation of over 20 large language models (LLMs), providing insights into the current state of LLMs’ medical capabilities.

**Weaknesses:**

1.	ClinBench focuses solely on diagnostic problems and adopts only a multiple-choice format, which limits its comprehensiveness in evaluating the full spectrum of medical reasoning and decision-making abilities.
2.	The key observations reported are either self-evident (e.g., reasoning models outperform non-reasoning ones) or based on unfair comparisons (e.g., comparing medical LLMs trained on older backbones with state-of-the-art general models to conclude that “medical LLMs of similar size have no clear advantage over general models.”) Moreover, these observations offer limited insights for future MedLLM development.
3.	Although ClinBench includes a rare-disease subset, it contains only 79 cases, which is relatively small compared to the 2,000+ cases in the full benchmark.

**Questions:**

1.	I noticed that all models achieved extremely large PPL values (10^120) on ClinBench in Appendix Table 8. This is highly unusual—could there be a computational or reporting error?
2.	Appendix Table 9 shows that about 10% of errors are caused by format issues. Given that existing LLMs generally possess strong instruction-following ability, this suggests there might be problems with the evaluation prompt or answer parsing. How does ClinBench handle answer extraction and parsing?
3.	In Figure 4 (right-side example), there is still textual description of imaging content (“The CT (Figure 1 [Lung CT]) scan showed multiple small cystic lucencies with clear borders in both lungs”) in the ClinBench_MM subset, which raises concerns about the data quality of ClinBench. How many undergraduate students annotated each case? Was any cross-validation performed among them? What was the average experience level (years of practice) of the experienced physicians involved? How many samples were manually inspected in total, and what was the pass rate?

---

### Official Review · Reviewer_1CXe · 2025-11-01

**Soundness:** 2
**Presentation:** 2
**Contribution:** 2
**Rating:** 2
**Confidence:** 5

**Summary:**

This paper introduces ClinBench, a benchmark built from real clinical cases published in medical journals. It also benchmarked large language models in these more clinically relevant medical questions.

**Strengths:**

1. The dataset is derived from authentic case reports, making the questions more difficult and closely aligned with real clinical reasoning. This gives the benchmark higher practical value.

2. Human experts are involved in both data construction and evaluation, which helps improve question quality and reduce ambiguity.

**Weaknesses:**

1. The dataset is not publicly available, so the results cannot be independently verified or reproduced. Moreover, the data construction process itself is not particularly novel, many existing datasets have also been rewritten from medical literature.

2. Using GPT-4o to generate distractor options may introduce bias into the dataset.

3. The paper lacks statistical analysis. Since experiments were repeated three times, confidence intervals and significance tests should be reported.

4. Lack of quality control of human annotation, such as agreement or iterative confirmation.

**Questions:**

1. GPT-4o is a closed-source model—how was perplexity (PPL) computed?


2. The paper is inconsistent about the number of specialties: is it 11 as in the tables or 12 as in the abstract?


3. What are the concrete differences between this dataset and previous ones built from medical literature?


4. How is “performance saturation” defined in this context, and how do Tables 1 and 2 support that claim?

---

### Note · Authors · 2026-01-06

**Comment:**

We would like to sincerely thank the reviewers and editors for their time, efforts, and constructive comments on our manuscript. Their valuable suggestions have provided us with important insights to further improve our work. We will take this opportunity to carefully revise the manuscript, improve the overall writing quality, and include additional analyses to strengthen and clarify our results.

**Withdrawal Confirmation:**

I have read and agree with the venue's withdrawal policy on behalf of myself and my co-authors.